# Regional Interactions in Social Responses to Extreme Climate Events: A Case Study of the North China Famine of 1876–1879

**Xianshuai Zhai [1,2], Xiuqi Fang [1,2] and Yun Su [1,2,*]**

[1]   Faculty of Geographical Science, Beijing Normal University, Beijing 100875, China;
     201831051021@mail.bnu.edu.cn (X.Z.); xfang@bnu.edu.cn (X.F.)
[2]   Key Laboratory of Environmental Change and Natural Disaster, Ministry of Education, Beijing Normal
     University, Beijing 100875, China
[*]   Correspondence: suyun@bnu.edu.cn

**Abstract:** The North China Famine of 1876–1879, known in Chinese as the Dingwu qihuang (丁戊奇荒), is a famous case of drought-induced famine in Chinese history. The purpose of this paper is to provide empirical and historical evidence for understanding the impacts of extreme climate events and major disasters and the mechanisms of adaptation. From the aspects of famine-related migration and the allocation of relief money and grain, the regional interactions in social responses to extreme climate events were analyzed. This paper collected 186 records from historical documents. Regarding the regions as the nodes and the relationships between regions as the links, the spatial patterns of famine-related migration and the allocation of money and grain from 1877 to 1878 were rebuilt. The results show that, firstly, famine-related migration appeared to be spontaneous and short-distanced, with the flow mainly spreading to the surrounding areas and towns. Secondly, as a state administrative action, the relief money and grain from the non-disaster areas were distributed to the disaster areas. However, the distribution of relief grain affected the equilibrium of the food market in non-disaster areas, which led to fluctuations in food prices.

**Keywords:** drought; regional interaction; North China Famine of 1876–1879

## 1. Introduction

Scholars have been studying the complex interactions between climate change and human history [1], and history is key to understanding the present and future. One of the major research themes of the Past Global Changes (PAGES) project focuses on the social impacts of historic extreme climate events and the responses, as well as the mechanisms and processes of past human-climate-ecosystem interactions at multiple spatial and temporal scales. It aims to enhance our understanding of the influence of contemporary climate change and the adaptation of human society. This is an international effort to coordinate and promote past global change research [2].

Among the studies on the impacts of and responses to the historic extreme climate events, most cases discussed the relationship between the human social system and the climate-environment system in the same region, but rarely involve regional correlations or common responses. In fact, when the impact of extreme climate events exceeds the regional carrying capacity, not only the affected areas, but also the initially non-affected areas can be influenced. There will be a common response in both affected areas and non-affected areas. For example, from 1813 to 1815, floods and droughts struck many countries of Europe, resulting in crop failures. Approximately 8000 refugees from Southern Germany migrated to Russia in the east. France, Italy, and the Netherlands imported food grains from Egypt, Russia, the United States, and some other regions [3]. In Australia, during the period

of 1800–1945, in the face of drought or floods, the social responses included the relocation of towns and the establishment of dams to coordinate water consumption in the upstream and downstream areas of the river basin [4]. In the southern part of North America, there were several severe drought events in the 9th to 14th centuries, and people there abandoned the infrastructures and migrated [5]. In China, from 1560 to 1890, it was at the height of the Little Ice Age that the climate fluctuated violently. The social responses, including famine-related migration and the allocation of money and grain, can be observed [6–12].

The presented case studies show that regional interaction has become an essential way of social response to historic extreme climate events. However, there is a lack of research on the characteristics, processes, and mechanisms of these regional interactions. Given the impact of extreme climate events and the social response could generate a complex multidirectional network in time and space [13], it is necessary to contribute to the research on the social response mechanism from the perspective of regional interaction.

In the present, global connectivity is continuously enhancing, and regional integration is deepening. It can be inferred that at the local, national, regional and global levels, the possibility of being affected by extreme events is increasing. According to the report of the American Meteorological Association, climate change is closely related to extreme events, and these events will seriously threaten the social economy and human life [14]. Therefore, inter-regional coordinated responses are urgently needed. IPCC's report pointed out that risk transfer and sharing will be an effective way of social response [15]; yet considering the interdependencies between regional economic and social systems, it may have opposite effects on different regions, which means the disaster risks could be either reduced or even amplified for a certain region involved [16,17]. Although it is impossible to reproduce the exact results of the response to the past events, the mechanisms, experience, and lessons of regional interactions in response to the historic extreme climate events are still be of an essential reference value.

Using documentary evidence to study past extreme climate events has become a recognized method [18], which emphasizes China's advantages in researching the social impacts of and response mechanisms to past climate change. On the one hand, the monsoon climate in China is characterized by its instability, and the traditional agriculture-based economy made the socio-economic system significantly sensitive to the changes in climate. On the other hand, China owns abundant and continuous documentary records left by its long history, such as historical books, local chronicles, archived documents, private diaries, etc. Besides, newspapers, which were first published in China in the early 19th century, are the documentary records with a high temporal resolution. They can not only be used to reconstruct the precipitation, temperature and other weather conditions in history [19,20], but also to explore the whole development process of historic extreme climate events within the socio-economic systems [18,21].

In this paper, focusing on the famine-related migration and the allocation of money and grain, the spatial and temporal features of the regional interactions in response to the North China Famine of 1876–1879 (known in Chinese as the "Dingwu qihuang" or the "Incredible Famine of 1877–1878") are analyzed. Combining with exploration on the after-effects of the famine on both disaster areas and non-disaster areas, the results provide the empirical evidence for understanding social response mechanisms from the perspective of inter-regional linkages.

## 2. Data Sources and Research Methods

### 2.1. Case Selection

North China is a region with a temperate monsoon climate, prone to drought in spring, summer and autumn. In the past 2000 years, there have been 227 extreme drought events in North China. Droughts occurred more frequently in 150–200 A.D., 550–800 A.D., 1050–1100 A.D., and 1850–1900 A.D. [22].

From 1876 to 1879, five provinces in North China, namely, Shanxi, Henan, Shaanxi, Hebei, and Shandong (Figure 1), suffered a severe drought. The reconstructed precipitation (wet/dry) series indicated that it was the most severe drought in this region in the past 300 years [23]. Sea surface temperature anomalies in the eastern Pacific region and intense El Niño events [24] had resulted in the weakening of the East Asian monsoon and precipitation variability, which were the direct causes of this drought event. It had global effects, with several regions experiencing extreme drought at the same time, including Australia, Europe, North America and South America [25–28].

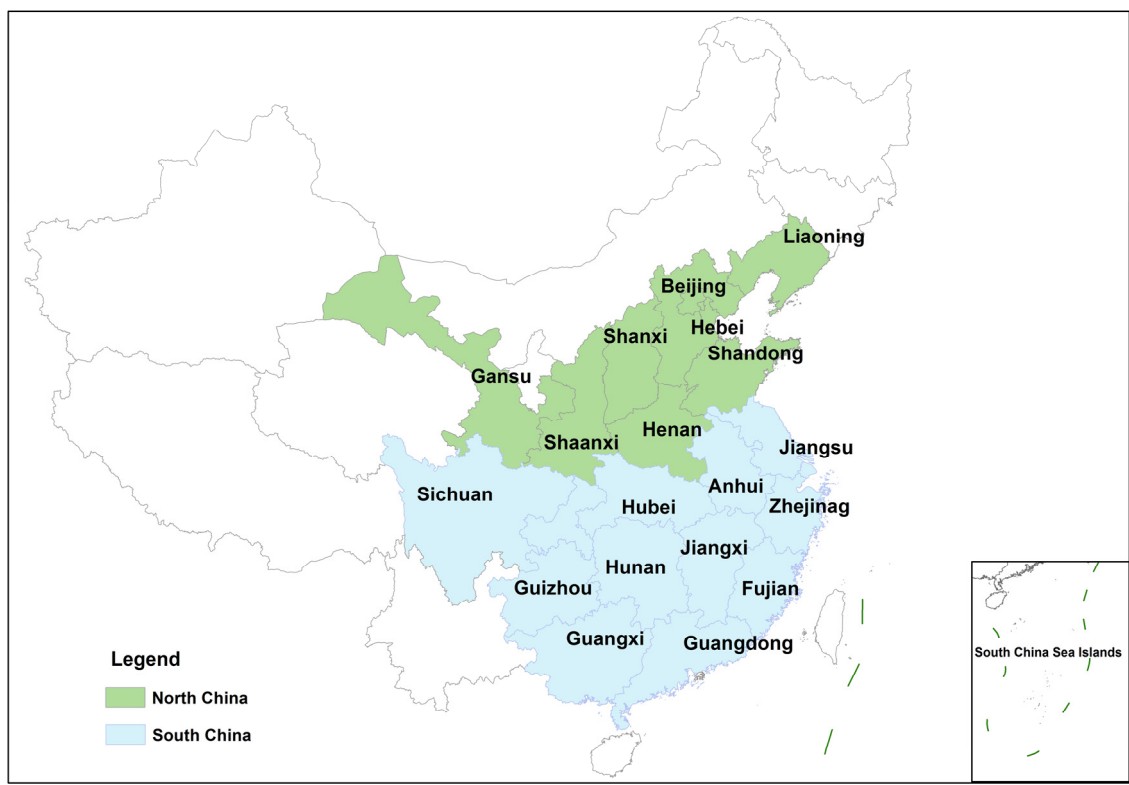

**Figure 1.** Study area in the paper. In the North China Plain: Beijing, Hebei, Shandong, and Henan; In the Loess Plateau: Shanxi, Shaanxi, and Gansu; In the Yangtze River Basin: Sichuan, Guizhou, Hubei, Hunan, Anhui, Jiangxi; Jiangsu, and Zhejiang; andIn the Jiangnan region: Fujian, Guangxi and Guangdong.

In China, the year 1877 was classified as a "Ding" year, and 1878 was classified as a "Wu" year. Because the worst period was in 1877–78, this extreme drought event was historically known as the "Dingwu qihuang". In 1877, 20% of the villages in Shanxi Province experienced harvest failures, and in the central Shaanxi Province, the harvest rate of grains during the fall harvest season was merely 30%. The famine reached its peak in 1877, with Shanxi and Henan worst affected [29]. Worse still, an epidemic occurred soon after and had spread over a large area during the spring and summer of 1878.

Regarding the social factors, frequent warfare in the late Qing Dynasty, fiscal crisis, and overburdened tenants aggravated the severity of disaster [30–32], leading to severe damage to productivity, homelessness, and social crisis [23].

In the end, approximately 160 to 200 million people were affected by the drought, and about 9.5 to 13 million people died from famine and disease. Many worst-hit counties in Shanxi and Henan provinces had lost over 50% of their population, with the death toll passed 5 million and 1.8 million respectively [33].

## 2.2. Study Area

The study area was divided into two parts (Figure 1). One is the affected areas. They are located mainly in the Loess Plateau and the North China Plain, which are the major wheat-growing areas in China, with a long development history and large population, containing the above-mentioned five drought-stricken provinces. The capital city and the political center of China, Beijing, is also in this region. This region is the target of the relief efforts carried out by the Qing government. The Qing state's responses to the famine consisted of a variety of strategies, such as allocating relief silver and grain and reducing or canceling taxes.

The other part is the south region, containing the Yangtze River Basin and Jiangnan region. It is the place where the economic center of China in the Qing Dynasty was located, and the resources for the disaster relief mainly came from. The landforms of this region are featured with the plain area along the middle–lower reaches of the Yangtze River, and the hilly area in the southeastern part. Different from the north region, this area is with a subtropical monsoon climate. Good hydrothermal conditions and well-developed water systems are conducive to the growth of rice, wheat, and a variety of cash crops, as well as the development of forestry and fishing.

## 2.3. Data Sources

The data about the North China Famine of 1876–1879 were extracted from Qing Shi Lu [34], Shenbao [35], Disaster annals in recent China [36], and Qing Tong Jian [37] (Table 1).

Qing Shi Lu is a long-term compilation of the chronicles of the Qing dynasty. It contains a total of 4363 volumes. The materials in Qing Shi Lu are originally from the official documents of the Imperial Cabinet and other departments, the pieces of writing from the National Historical Archives, and some first-hand materials such as the emperor's anthology and handwriting [10]. The historical materials in Qing Shi Lu are of exceptionally high value.

By 1876, the Shenbao had established itself as a commercially successful newspaper that carried the only public and serious discussion of many public issues in China [38]. From 1876–79, Shenbao's critical coverage of the famine focused not only on the five hardest-hit northern provinces but also on some other areas influenced by the drought event [39].

Disaster annals in recent China systematically and chronologically expounded on the natural disasters in China from 1840 to 1919, combined with explicit analyses on the time, location, extent, causes and social influences of various natural disasters, as well as the effectiveness and gaps of disaster mitigation measures [40]. The data is of high quality and reliability.

In addition to the above sources, data source about Dingwu qihuang is also supplemented by Qing Tong Jian. Its collection of historical materials is complete and reliable, and it discusses in detail politics, society, finance, economy, transportation, war, etc.

**Table 1.** Information on sources of the data about the North China Famine of 1876–1879.

| Data Sources | Temporal Coverage | Language | Access |
|---|---|---|---|
| *Qing Shi Lu* | 1876–1879 | Chinese | ISBN 978710105626 |
| *Shenbao* | 1876–1879 | Chinese | https://www.neohytung.com/ |
| *Disaster annals in recent China* | 1876–1879 | Chinese | ISBN 9787535510839 |
| *Qing Tong Jian* | 1876–1879 | Chinese | ISBN 9787203039075 |
| *Food Price Database in the Qing Dynasty* | 1876–1879 | Chinese | http://mhdb.mh.sinica.edu.tw/foodprice/ |
| *Zhang's research* | 1877–1878 | English | doi:10.3724/SP.J.1248.2010.00091 |

After the removal of redundant records, a total of 186 historical records were extracted and classified (Table 2). We excerpted 96 records from Qing Shi Lu, 70 from Shenbao, 13 from Disaster annals in recent China, and 7 from Qing Tong Jian. Figure 2 shows that 1877–1878 is the key period, with significantly more records identified for the famine event.

**Table 2.** Classification of the historical records for the North China Famine of 1876–1879.

| Contents | Number of Records | Records Indicated Directions | Records Indicated Quantities |
|---|---|---|---|
| Famine-related migration | 56 | 30 | 6 |
| Allocation of money and grain | 74 | 74 | 34 |
| Social unrest | 56 | 56 | 7 |
| **Total** | **186** | **160** | **47** |

| Contents/Years | 1876 | 1877 | 1878 | 1879 |
|---|---|---|---|---|
| Famine-related migration | | | | |
| Allocation of money and grain | | | | |
| Social unrest | | | | |

No data　≤ 3%　3-4%　4-5%　5-7%　7-10%

**Figure 2.** Percentage of days with records in each year from 1876–1879.

This paper selected data of wheat prices from the Food Price Database in the Qing Dynasty [41] (Table 1). The database is based on historical documents in the First Historical Archives of China and the Institute of Economics of the Chinese Academy of Social Sciences.

The data on the famine-struck area and plague-infested area is from Zhang's research [42] (Table 1), which was based on climate records extracted from historical documents in "A compendium of Chinese meteorological records of the last 3000 Years (in Chinese)" [43].

*2.4. Information Extraction*

We extracted information on key activities, noting down the general processes, times and places of occurrence, and the numbers of starving migrants, or the amount of relief silver and grain. The times in the documents were converted into Gregorian calendar time format with monthly temporal resolution. The locations were recorded based on the current provincial-level and prefecture-level administrative divisions in China.

In the Qing Dynasty, grain was measured in piculs ("dan" in Chinese), and silver used in monetary transactions was measured in taels ("liang"). In the 19th century, 1 picul of rice weighed 60 kg [29]. 1 tael of silver was equal to $29.6 [44], converted according to the purchasing power of silver at that time.

Firstly, we identified the records of famine-related migration, for example, "(1878) this year, there was a severe drought in Henan, and starving people moved to Xuzhou in search of food [35]" (Figure 3a). Then the migration route was noted down: Henan→Xuzhou.

Secondly, the records of silver and grain allocation were also extracted. "(1877) when the severe drought struck Shanxi and Henan, the central government allocated 280,000 taels of silver to Shanxi and 120,000 taels of silver to Henan. Equally, 40,000 piculs of grain from the granaries in Anhui and Jiangsu shall be allocated to Shanxi [34]" (Figure 3b). The silver allocation was noted down: "Beijing→Shanxi, 280,000 taels; Beijing→Henan, 120,000 taels". The grain allocation: "Jiangsu/Anhui→Shanxi, 40,000 piculs". The grain in the paper refers to wheat and rice, the two major food staples in China.

Finally came to the records of social unrest during the famine. It includes revolt, banditry, and insurrection, e.g., "(1878) the bandits rebelled in Shanzhou [34]" (Figure 3c). In this record, the site of the unrest event was the Shanzhou district, which is now the Sanmenxia city of Henan Province.

(**a**)

(**b**)

**Figure 3.** *Cont.*

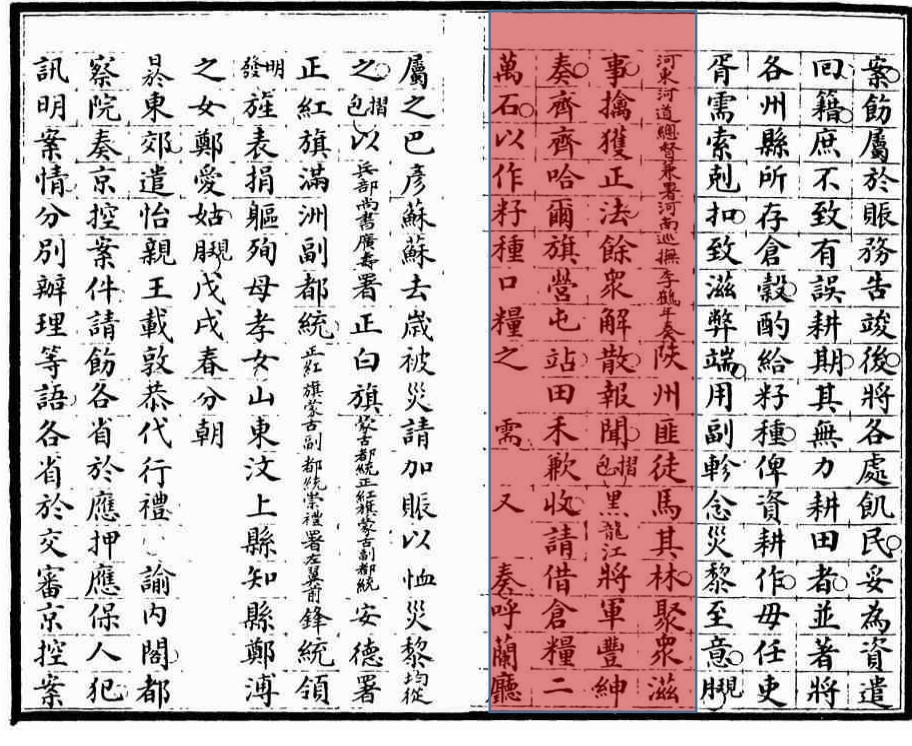

(**c**)

**Figure 3.** Historical records related to the North China Famine of 1876–1879 (**a**) is from Shenbao, about the famine-related migration; (**b**) is from Qing Shi Lu, about the allocation of money and grain; and (**c**) is also from Qing Shi Lu, about social unrest.

*2.5. Spatial Analysis*

To demonstrate the regional interactions in the social responses, the origins and destinations of the starving migrants were regarded as nodes, and the migration flows were regarded as the links. A spatial network of the famine-related migration in 1877–1878 was built (Figure 4). Similarly, taking the provinces where silver and grain transferred out and in as the nodes, the inter-provincial transfers as the links, and the number of transfers as the weight, a weighted network of money and grain allocation in 1877–1878 was built (Figure 5). In the networks, the degree refers to the number of direct links connected with each node. It can reflect the number or range of connections between different areas. Using ArcGIS, the lengths (distances) of migration and allocation flows can be calculated.

Besides, regarding the sites where the social unrest events took place as the core, point density analysis was conducted in ArcGIS. In the analysis, a particular sample point is set as the center, then a neighborhood is defined around each center with a certain value of search radius. After that, points of different weight values that fall within the neighborhood are identified. Points are weighted higher if they are closer to the centers. The weights gradually reduce until they drop to zero at the edge of the neighborhood. The density value is the sum of the values of the sample points divided by the area of the neighborhood [45,46]. In this paper, the unrest occurrence sites are the centers, and the search radius is 100 km. The unrest density value is divided into eight levels according to the number of unrest events per 10,000 square kilometers. The frequency of occurrence and the spillover effects of social unrest were analyzed.

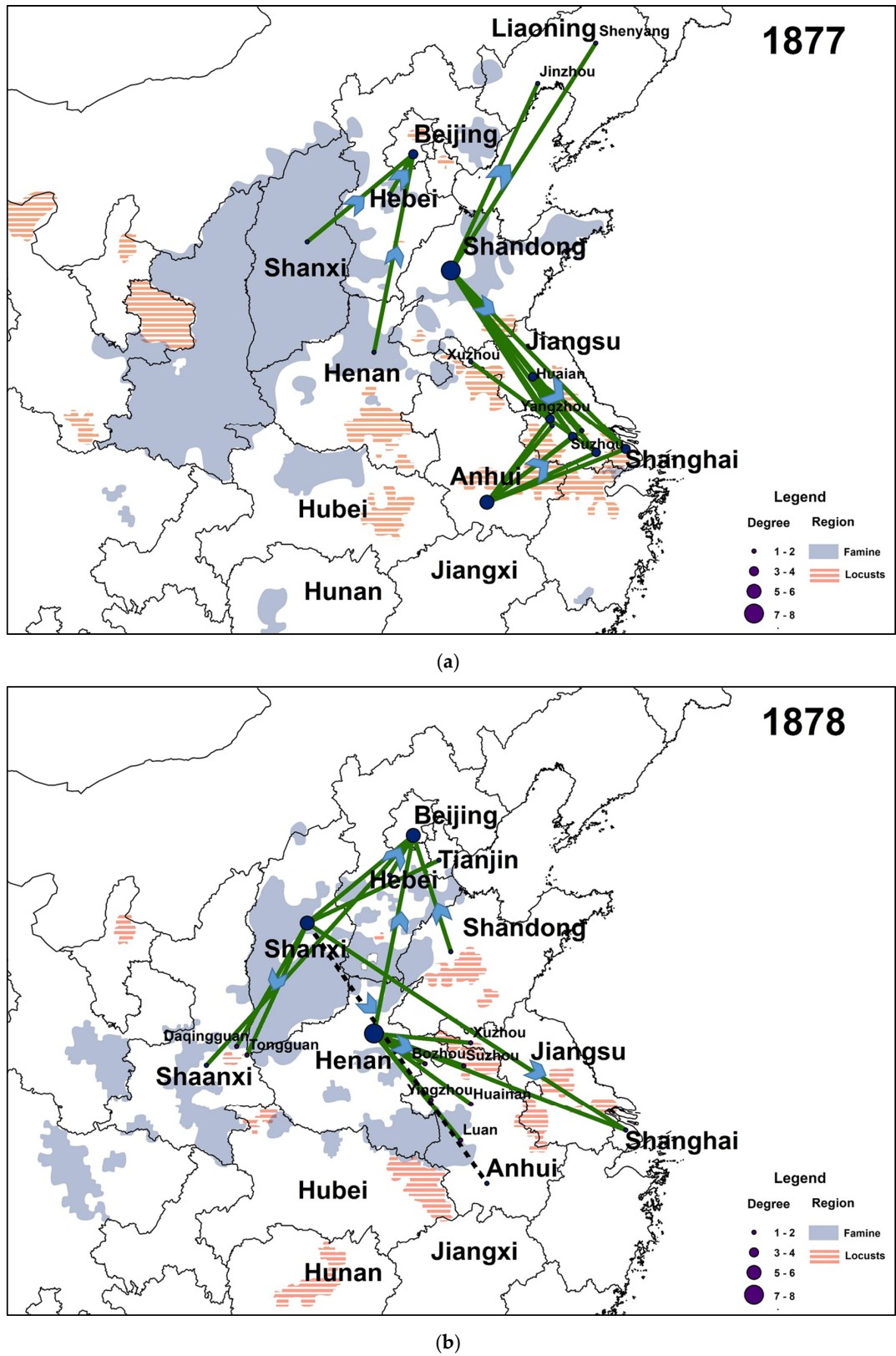

**Figure 4.** (**a**) Origins, destinations and sizes of the famine-related migration in 1877. (**b**) Origins, destinations and sizes of the famine-related migration in 1878. Straight line: migration flows with exact locations of destinations; Dotted line: migration flows with approximate locations of destinations; Degree: the number of direct links connected with a node.

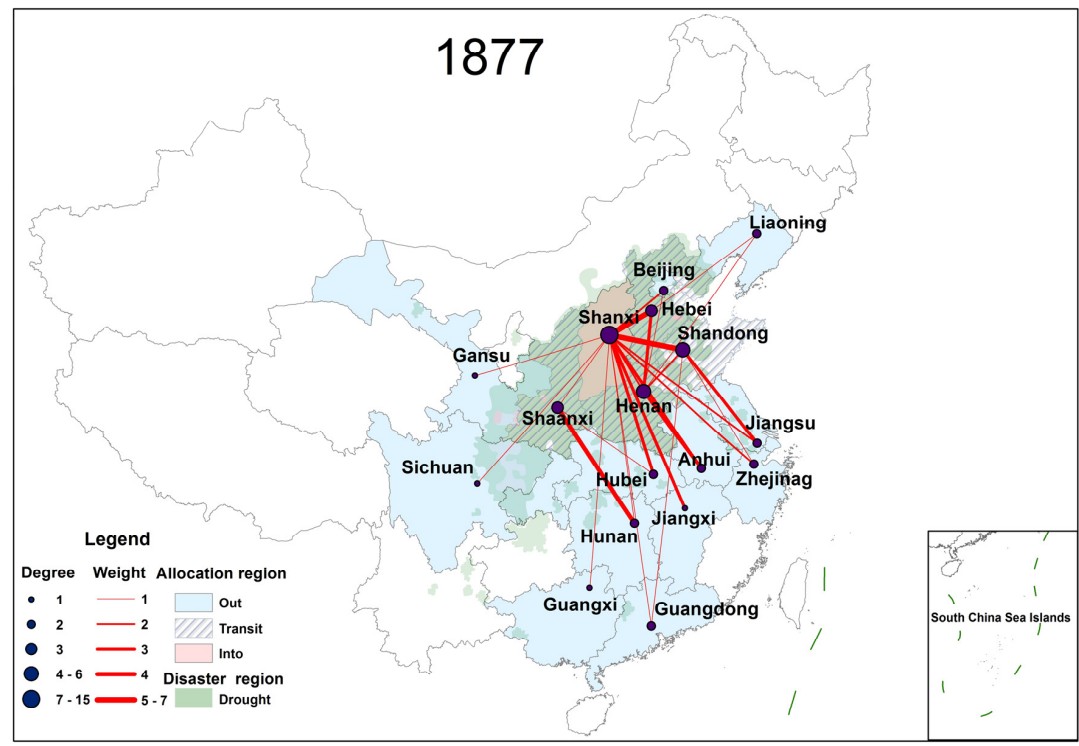

(**a**)

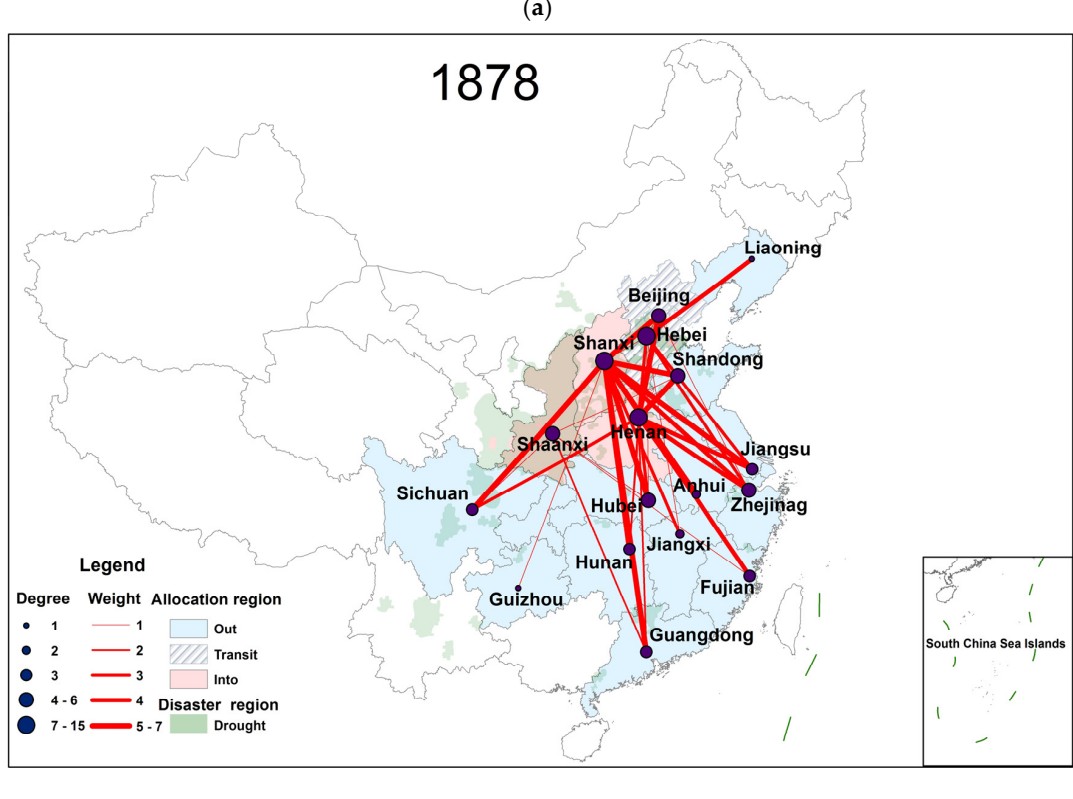

(**b**)

**Figure 5.** (**a**) Spatial characteristics of grain and monetary allocations in 1877. (**b**) Spatial characteristics of grain and monetary allocations in 1878. Degree: the number of direct links connected with a node; Weight: the number of inter-provincial transfers.

## 3. Results

### 3.1. Regional Interactive Response and Characteristics

#### 3.1.1. Spatial Characteristics of Famine-Related Migrations

Historically, Chinese farmers have had a tied connection to their homeland and farmland. Large-scale population migration is temporary and spontaneous in the period of extreme climate events. The migration is often driven by famine, plague and other events triggered by major meteorological disasters [6]. The transition from being starving to homeless is the result of the local social system losing its ability to adapt and the failure of individual survival strategies.

Famine victims tended to migrate from the hardest-hit areas to the nearest slightly-impacted areas, and non-disaster areas (Figure 4). In 1877, migrants left from Shandong and Anhui to the south of Jiangsu. Some people left from Shandong to Liaoning, while others moved from Shanxi, Henan, and Hebei to Beijing. The victims migrated between 130 and 766 km (straight line distance, the same below), with an average migration distance of approximately 429 km. In 1878, famished people in Henan moved to Beijing and Anhui. Some people in Shandong, Hebei, and Shaanxi moved to Beijing, and others in Shanxi moved to Hebei, Beijing and Anhui, as well as to the Daqingguan and Tongguan in Shaanxi. Famine victims migrated between 130 and 1080 km, with an average migration distance of approximately 460 km.

In this historical period, limited by poor traffic conditions and the physical weakness, the spontaneous famine-related migration flows just spread from the disaster areas to the nearby areas. In 1877–1878, the harvest rate of grains in Anhui was only about 50%, while in Jiangxi, Hubei, and Hunan, which are farther away from the disaster areas, the harvest rate reached over 60% or 70% (Table 3). Although with relatively low harvest rate, Anhui was still one of the major destinations in the famine-related migration due to its proximity to the disaster areas. Meanwhile, the migration directions of the famished people appeared to be relatively fixed and stable, indicating the influences of regional politics, dissemination of the relief information, and customs and traditions [47].

**Table 3.** Harvest rate of grains in different provinces [48].

| Year | Anhui | | Jiangxi | | Hubei | | Hunan | |
|------|-------------------|-------------------|-------------------|-------------------|-------------------|-------------------|-------------------|-------------------|
| | Summer Harvest | Autumn Harvest | Summer Harvest | Autumn Harvest | Summer Harvest | Autumn Harvest | Summer Harvest | Autumn Harvest |
| 1877 | 50%+ | 50%+ | 70%+ | 70%+ | 60%+ | 50%+ | 60%+ | 70%+ |
| 1878 | 50%+ | 50%+ | 60%+ | 60%+ | 60%+ | 60%+ | 60%+ | 70%+ |

There was a visible difference in the economic conditions between towns and villages in China during the Qing dynasties. Under the circumstance of the rural economic decline, cities and towns became the destinations of the migrants. Soup kitchens and shelters there offered more chances of survival. For example, the capital city at that time, Beijing, is close to the affected provinces, and it had become a popular destination for the starving migrants. Besides, some other places attracted migrants due to their advantageous geographical location and sophisticated traffic system. For instance, in 1878, the whole area of Shaanxi Province suffered a severe drought, but its two towns, Tongguan and Daqingguan, had experienced the entry of famished people from the Shanxi Province on the east. Lying close to the shared borders of Shaanxi, Shanxi and Henan provinces, these two towns were distribution centers of relief goods and had been of strategic importance since ancient times.

#### 3.1.2. Spatial Characteristics of Money and Grain Allocations

In China, a state unified by centralized political power, the allocation of money and grain is a state administrative action. It is normal that the central government provides emergency financial assistance and food aid to disaster areas. The relief silver and grain are mainly from state banks

and granaries, which are supplied by every locality on a regular basis. Besides, some resources in the non-disaster areas could be requisitioned as emergency relief by the central government for the duration of the famine.

In 1877, a total of 17 provinces were involved in the allocation of money and grain (Figure 5). The main drought-stricken areas, Shanxi, Henan, Shandong, Shaanxi, and Hebei, received the most amount of silver and grain. From the perspective of spatial pattern, the allocation is featured with a core-ring structure: (1) Shanxi, lying in the core, received the most amount of silver and grain; (2) Shandong, Henan, and Hebei, lying on the second layer, received the relief from other provinces and also supplied resources for Shanxi; (3) Hunan, Anhui, Jiangxi, Jiangsu and other provinces in the Middle–Lower Yangtze Plain, lying on the most peripheral ring, experienced mainly the outflow of silver and grain. In 1877, the straight-line distances of silver and grain allocations were between 270 and 1635 km, and the average transfer distance was approximately 800 km.

In 1878, also 17 provinces were involved in the relief allocations, but the spatial structure appeared to be a complex network (Figure 5). The amounts of the relief grain being allocated to Shanxi, Henan and Hebei accounted respectively for 34%, 39% and 27% of the total number of grain transfers. The distribution appeared to be more balanced than that in 1877. Shanxi, Shaanxi, Henan, and Hebei were the main receivers of the relief silver, accounting for 96% of the total silver transfers. Zhejiang, Hubei, and Hunan in the Yangtze River Basin remained as the suppliers. Apart from them, Fujian, Guangdong and Sichuan in the farther south also became the main relief suppliers. In 1878, the straight-line distances of silver and grain allocations were 132–1635 km, with an average of approximately 860 km.

Unlike the spontaneous migrations of famine victims, the allocation of money and grain is government action. It was at a larger spatial scale and with more frequent transfers. Moreover, the spatial pattern of the money and grain allocations was relatively more complex, as it would vary according to the severity of the disasters and national relief policies.

### 3.1.3. Temporal Characteristics of Famine-Related Migrations and Money and Grain Allocations

The records of the money and grain allocations mainly came from official documents, so the time of the documents was the time of the allocations. While the records of famine-related migrations might only appear when there were many migrations, they can still roughly indicate the period of mass famine-related migration.

According to the records that contain clear time information of the events in 1877–1878 (Figure 6), the relief allocations and famine-related migrations appeared to be seasonal and temporary, and shared with a similar peak period, which was from October 1877 to May 1878. However, the duration of money and grain allocations was longer than that of famine-related migrations.

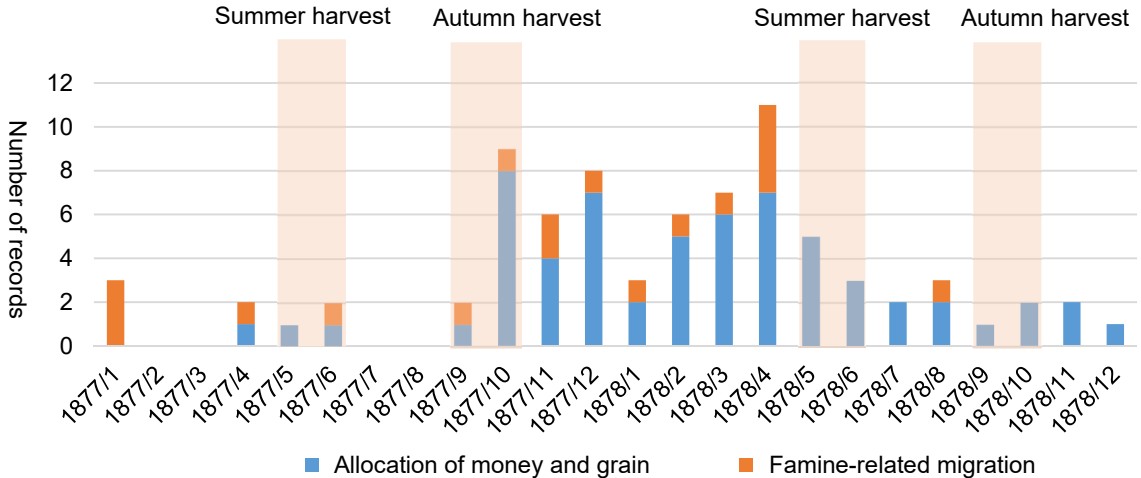

**Figure 6.** The number of the records of famine-related migrations and relief allocations in 1877–1878.

In North China, during the Qing dynasty, the summer harvest season started from May to June, and the autumn harvest season was from September to October. For example, in Shandong Province, wheat was harvested in the fifth lunar month (June in the Gregorian calendar), while sorghum and millet were harvested in the eighth lunar month (September in the Gregorian calendar) [49].

A small number of records of migrations before the summer harvest in 1877 indicated the occurrence of drought, but it also suggested that the situation was not beyond control. However, a large number of records were found of the time period from October 1877 to May 1878. The autumn of 1877 experienced a severe harvest failure as the consequence of persistent extreme droughts in the previous months. In the spring and summer of 1878, as the secondary disaster, plagues began to spread in the disaster areas. These two might be the main causes of the significant rise of the number of records, which suggested that the impacts of the extreme droughts had exceeded the response capability of a single region. In this stage, regional interactions were necessary for the mitigation of disaster impacts.

## 4. Discussion

### 4.1. Influence of Money and Grain Allocations on Regional Food Prices

Food prices directly reflect the food supply and demand, and are negatively correlated with the crop yields, and are also the indicators of social stability [50]. Therefore, variations in food prices in disaster areas and non-disaster areas can indicate the effects of regional interactions and coordinated responses to the extreme drought events. Wheat prices from 1876 to 1879 were selected from the Food Price Database in the Qing Dynasty [41] for the analysis, as wheat has always been the primary food staple in China. Figure 7 shows the lowest and highest wheat prices in different provinces and regions in each year from 1876 to 1879.

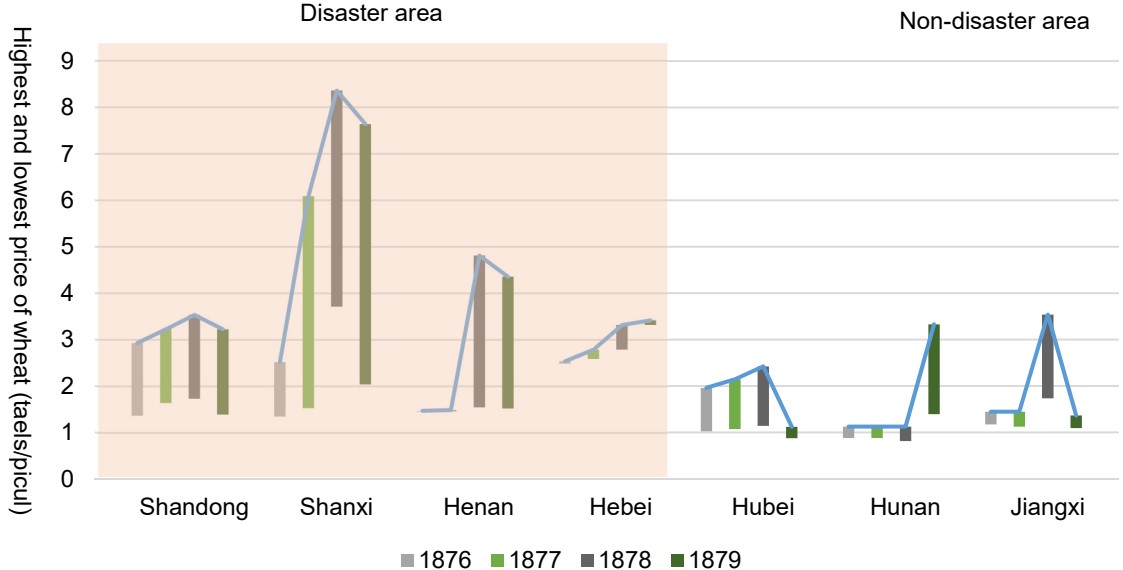

**Figure 7.** Provincial wheat prices in 1876–1879. Blue lines: variations of the highest annual wheat price in the province.

The wheat prices in every province of the disaster area had risen with fluctuations. In Shanxi and Henan, the prices in 1877 and 1878 could be over four or five times higher than usual. Wheat prices in Shanxi fluctuated most violently, and the extent of the rise was the most significant. In 1878 and 1879, wheat prices in Henan soared. In Shandong and Hebei, however, the wheat prices were relatively stable, with just small increases. A positive correlation can be observed between wheat price variations and the severity of droughts and famines.

The relief silver and grain alleviated some of the damages caused by the famine, but the effects were still insufficient. Taking Shanxi as an example, from 1876 to 1879, Shanxi had received 1.76 million

piculs of grain and 13 million taels of silver in total. The relief silver could purchase about 2.6 million piculs of wheat according to the average annual wheat price (5 taels of silver per picul) in Shanxi at that time. That is to say, on average, about 1.09 million piculs of wheat were distributed to Shanxi each year. This amount of wheat could support 650,000 famine victims to survive for 8 months after the autumn harvest failure till the next summer harvest, based on the average daily food consumption of approximately 0.007 piculs per person [29]. However, at that time, the number of famine victims in Shanxi had exceeded 4 million [51], which means only 16% of the famished people could be supported. The situation remained severe, and some starving people even started resorting to cannibalism for survival. According to the records, starvation cannibalism occurred in respectively 42 worst-hit counties of Shanxi, 21 in Henan, and 11 Shaanxi [52].

In the non-disaster area, there were also obvious fluctuations in wheat prices. During 1877–1878, the harvest rates in Hubei, Hunan, and Jiangxi were just around 60–70% (Table 3). Sending relief grain to other provinces caused insufficient domestic supply, resulting in the sharp increases in wheat prices in Hunan and Jiangxi. On the one hand, inter-regional grain transfers worked effectively in stabilizing food prices in the disaster area. On the other hand, to some extent, the grain transfers disturbed the conditions of the food markets in the non-disaster area.

### 4.2. Influence of Famine-Related Migrations on Regional Social Stability

The regional interactions in social responses to the famine also profoundly influenced social stability. In 1877, more social unrest events ($\geq$2–4/10,000 km$^2$) took place in Henan, Shandong, Hebei, and Beijing (Figure 8), which were the main origins and destinations of the starving migrants. Very high unrest events density ($\geq$6–7/10,000 km$^2$) was found in Hebei. There were many reports of banditry and food robbery at the junction of Shaanxi, Shanxi, and Henan provinces. In 1878, the popular destination of famine migrants changed from the southern Jiangsu to Anhui, which made the social order of the former begin to improve. The places with a high density of social unrest events ($\geq$2–4/10,000 km$^2$) were recognized in Henan.

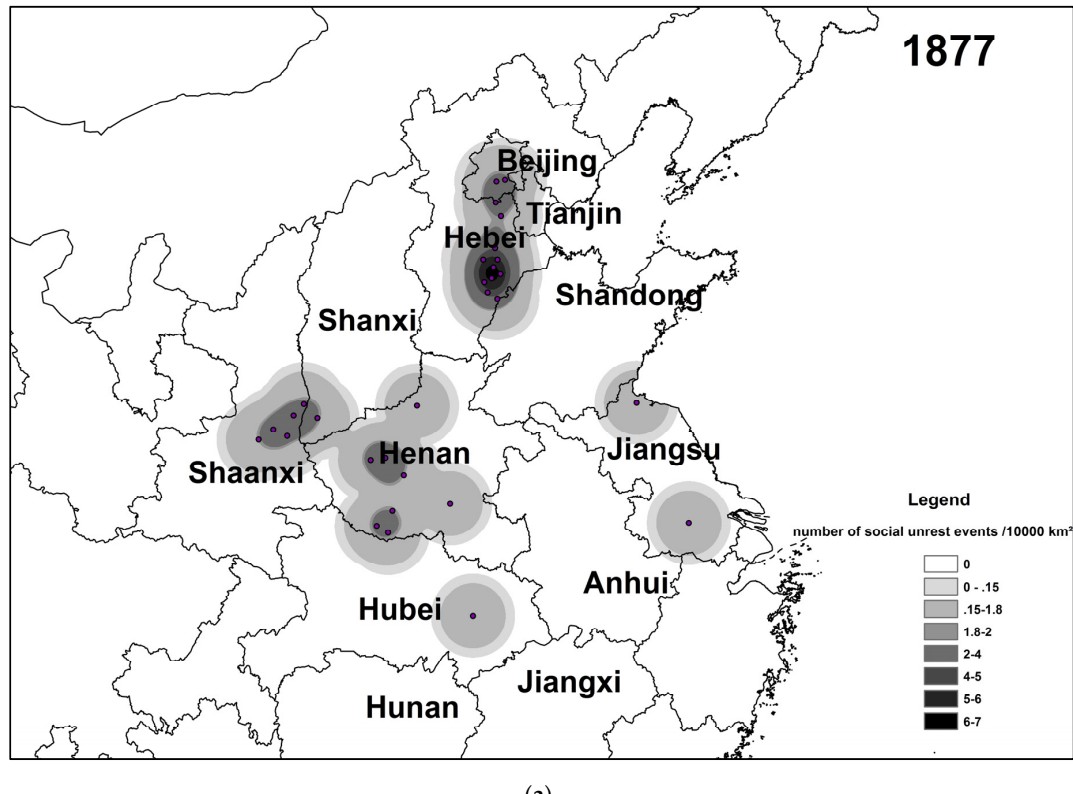

(**a**)

**Figure 8.** *Cont.*

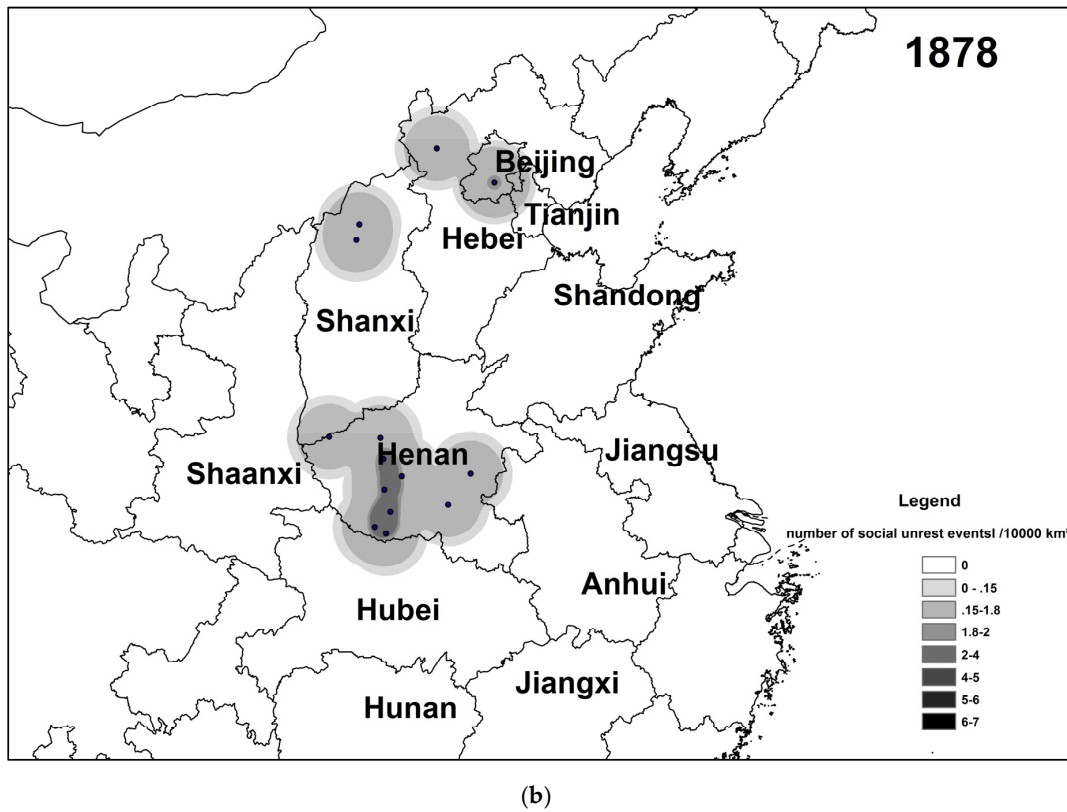

**Figure 8.** (**a**) Social unrest events density in 1877. (**b**) Social unrest events density in 1878.

When the famine victims left from the worst-hit areas to the surrounding slightly-impacted areas, or from villages to towns and cities, the social unrest events also "followed" with them from the disaster areas to the destinations for migrants. Thus, for the non-disaster areas, the management of the starving migrants was a challenge of regional governance.

### 4.3. Regional Interaction Responses and Transfer-Dispersion of the Impacts of Extreme Weather Events

Famine-related migration represents the dispersion of the population pressure in the famine-struck areas (Figure 9). The spatial redistribution of the famine victims dispersed the population pressure in their places of origin, but increased the population pressure in their destinations. Famine-related migration is also connected to social unrest events and represents the dispersion of the social impacts of the extreme drought events.

The allocation of money and grain is an administrative action, with the purpose of adjusting the gaps in food production between disaster areas and non-disaster areas, and increase the opportunities to obtain food for the victims in the disaster areas. Affected by the allocation, food prices in some of the non-disaster areas also increased. To a certain extent, the allocation of money and grain played a role in transferring the social impacts of disasters and had positive effects on the post-disaster recovery (Figure 9).

In Chinese history, normally there were two areas of destination for the migration driven by the events related to climate change. One is the southeastern part of China, with warm and humid weather and fertile soils. The other is the arid and barren northwest. Migrating to the northwest seems to go against common sense, but it was because of the increase of conflicts between farmers and herders, and the increase of invasion of the northern nomads in the dry period [53]. Between 1876 and 1879, extreme droughts led to a server famine in North China, and many people migrated in search of food. However, in spite of the food production, the starving migrants selected their destinations using the location and proximity to their homeland as the determining factors. This is because that the famine-related migrations at that time were mainly temporary, and it was difficult for starving people

to complete long-distance travel. Compared with the climate-driven migration mentioned above, the migration during the North China Famine of 1876–1879 was a spontaneous social response to the extreme disaster events and the dispersion of short-term population pressure in the disaster areas.

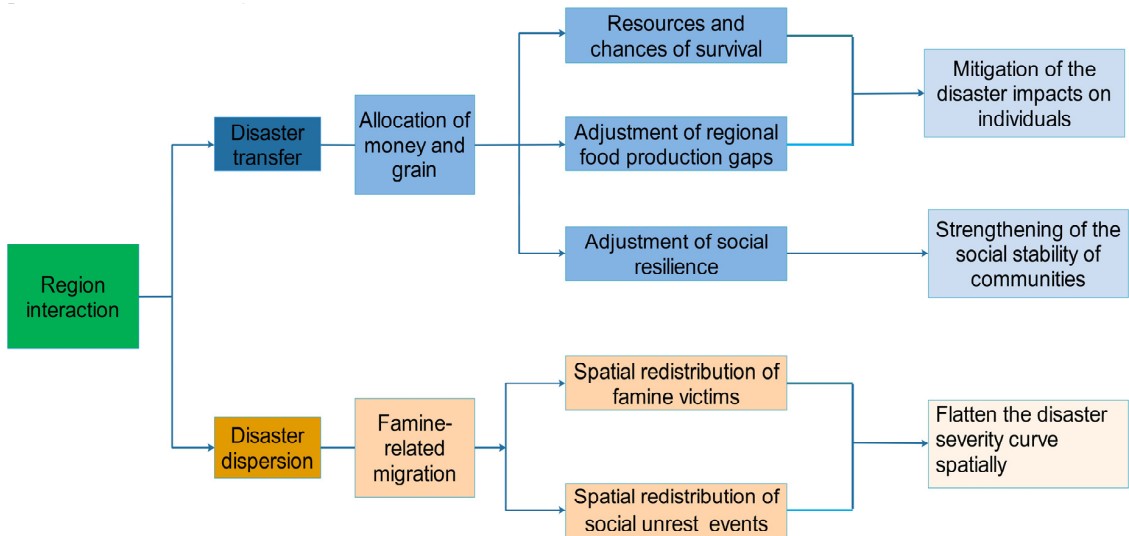

**Figure 9.** Process of regional interaction responses and transfer-dispersion of the impacts of disasters.

Based on the vast canal network, the central government planned, organized and carried out the allocation of money and grain, which was also the process of impact transfer across the whole country. In this network, the relationship between regions was more complex than that in the migration network. This time, in spite of the distances, the grain storage and accessibility of a place were the main factors in the decision-making process.

## 5. Conclusions

After the analysis of the spatial and temporal characteristics of famine-related migration and allocation of the disaster relief in the North China Famine of 1876–1879, the conclusions were summarized as follows:

(1)　Famine caused by extreme drought events was the main driving force of the migration. Famine-related migration was spontaneous and short-distanced, flowing into the surrounding towns and cities. The straight-line travel distances of most migrations were approximately 400 km. Famine-related migration spatially dispersed the population pressure but caused the spillover of social unrest.

(2)　As a government action, the relief silver and grain from the non-disaster areas were distributed to the disaster areas, with an average relief transfer distance of over 800 km. The transfers of the famine relief formed a complex spatial network. During the worst period of the famine, due to harvest failures, wheat prices were over four or five times higher than usual. The allocation of money and grain relieved some pressure on the food supply in the disaster areas but did not fundamentally change the situation. It also affected the equilibrium of the food market in the non-disaster areas, which led to the fluctuations in wheat prices.

(3)　The regional interactions in the process of responding to extreme climate events is a process of dispersion and transfer of the disaster events' impacts, which will have different risk effects to both disaster areas and non-disaster areas. In the context of increasing globalization and regional linkages, a higher capacity for integrated risk prevention and comprehensive administrative governance is required.

**Author Contributions:** X.Z. performed research, analyzed data, and wrote the original manuscript; X.F. revised the original manuscript of the paper and made contributions to the research ideas. Y.S. designed research, revised the original manuscript of the paper and discussed the results. All authors have read and agreed to the published version of the manuscript.

**Funding:** This research was funded by National Key Research and Development Program of China (No. 2018YFA0605602) and National Natural Sciences Foundation of China (No. 41771572).

**Conflicts of Interest:** The authors declare no conflict of interest.

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
