# Peer review of "Regional Interactions in Social Responses to Extreme Climate Events: A Case Study of the North China Famine of 1876–1879"

_atmosphere, doi:10.3390/atmos11040393_

Round 1
Reviewer 1 Report
Review of 'Regional Interactions of Social Responses to Extreme Climate Events: A Case Study of the Dingwu Famine (1877-1878 AD) in China' by Zhai et al.
In this work Zhai et al. try to provide some insight on the social responses to a well-known drought event in China. For it, they use existing documentary records, discuss some ideas on harvest prices and migrations.
My first concern with the work is about the scope of the journal. Although this is something to be assessed by the editorial board, I do not think that 'Atmosphere' is the right journal for this work. Indeed, the research presented has little to do with atmospheric sciences. Maybe, the authors or the board should consider moving the work to a different journal.
That said, I like the work. However, I consider that its presentation and discussion needs to be significantly improved before it can be accepted for publication. I list below some significant problems that need to be fixed before checking more in deep the work performed.
- Incomplete discussion of the drought event: The Introduction lacks a proper discussion on the existing literature on this well-know drought event. This section should be completed discussing results from previous research on the topic and citing such works. I link here three papers, but encourage the authors to go beyond my suggestions:
- https://doi.org/10.1371/journal.pone.0148072
- https://link.springer.com/article/10.1007/s10113-011-0245-6
- https://doi.org/10.1016/j.quaint.2012.03.011
- Incomplete captions: in general the captions of the figures are too short, without enough information to understand them. Captions in figures must be self-descriptive.
- The information provided on the Chinese names and regions must be improved. Probably, most of the readers will not be familiar with the names of Chinese provinces or its geopolitical distribution. Therefore, some extra background needs to be provided on this.
- The information on the documentary data sources must be presented better to make it clearer. I recommend a table listing the documentary sources, temporal coverage, original language, and how to gain access to them (availability, library, an online source, etc.).
Also, using newspapers is not so common. I suggest to include a few examples of similar studies and discuss them in the Introduction. Moreover, it is necessary to include a figure with the temporal coverage and gaps of the recovered records, such as suggested in the following works:
* https://doi.org/10.5194/cp-2-137-2006
* https://doi.org/10.1371/journal.pone.0039281
* https://doi.org/10.1002/wea.2841
- It would be great to include some illustrative photographies of the data sources used in this study.
- This study is relevant in the framework of extreme weather events. It would be good to include in the Introduction a specific paragraph on the evolution of extreme weather events, and specifically, droughts in the region studied. Beyond a simple citation to the IPCC, I recommend to include some discussion and citations on the impacts and recurrence of such events and the role of climate change. I would suggest using some of the results from the reports by the American Meteorological Society in the Bull. Am. Meteor. Soc.:
https://www.ametsoc.org/ams/index.cfm/publications/bulletin-of-the-american-meteorological-society-bams/explaining-extreme-events-from-a-climate-perspective/
- 'Grain' is too generic. Is it possible to clarify the cereal/crop?
Reviewer 2 Report
This is a very interesting research to focus on one famine event of national impacts. Furthermore, the study has considered the both political and individual responses to famines. It will add more knowledge on past social resilience to disasters. I only have some minor comments to improve the manuscript.
First, the figure legends are not informative enough. The readers may need more information to understand the results as shown in figures.
Second, the data adopted in the study is from the past records. I suggest the authors to give some examples of original records. Then, the readers know more how the authors quantified the historical records.
Third, Dingwu Famine is a very important case in Chinese history. Could authors explain more why they choose this case but not others?
Fourth, there are some studies of a long term, such as Pei et al (2019). What is the main innovative finding in the study? In fact, the case-based study is quite good practice in addition to existing long-term one. I suggest the authors to discuss more on their innovative points.
Fifth, as a minor suggestion, the language should be improve. The expression sometimes are vague and obscure.
Reference
Q. Pei, Z. Nowak, G. Li, C. Xu, W. K. Chan, The Strange Flight of the Peacock: Farmers’ atypical northwesterly migration from central China, 200BC-1400AD. Annals of the Association of American Geographers 109, 1583-1596 (2019).
Reviewer 3 Report
The topic of this paper falls with that of Atmosphere. It is academically sound, well designed, and well organised. It merits publication in Atmosphere after a proofreading of language.
Round 2
Reviewer 1 Report
Review of 'Regional Interactions of Social Responses to Extreme Climate Events: A Case Study of the Dingwu Famine (1877-1878 AD) in China' by Zhai et al. (atmosphere-759646 )
In this new version of their manuscript, the authors have addressed several of my previous comments, and apparently, greatly improved the work. The work is now better referenced, more complete, etc.
Unfortunately, the English language is bad and prevents to carry on a proper analysis of the work. In its current form, I can not perform a more deep review of the manuscript and its contents.
Therefore, I recommend 'Major revisions' and a new submission after performing heavy language editing. I have found mistakes, misspells, and a good number of grammar constructions are weird and near to impossible to understand.
I would like to point out that the concept 'extreme climate event' used by the authors is wrong in the context of this work. They mean 'extreme weather events'. Climate and weather are very different things.
Reviewer 2 Report
I have no comments. Thank you for your revision according to my comments of last round.